# The role of the MAD2-TLR4-MyD88 axis in paclitaxel resistance in ovarian cancer

**Mark Bates**[1,2,3,4]*, **Cathy D. Spillane**[1,2,3], **Michael F. Gallagher**[1,2,3], **Amanda McCann**[5], **Cara Martin**[1,2,3,6], **Gordon Blackshields**[2,3,6], **Helen Keegan**[1,2,3,6], **Luke Gubbins**[5], **Robert Brooks**[7], **Doug Brooks**[7], **Stavros Selemidis**[8], **Sharon O'Toole**[1,2,3,4‡], **John J. O'Leary**[1,2,3,6‡]

**1** Department of Histopathology, Trinity College Dublin, Dublin, Ireland, **2** Emer Casey Molecular Pathology Research Laboratory, Coombe Women & Infants University Hospital, Dublin, Ireland, **3** Trinity St James's Cancer Institute, Dublin, Ireland, **4** Department of Obstetrics and Gynaecology, Trinity College Dublin, Dublin, Ireland, **5** College of Health Sciences, University College Dublin, Belfield, Dublin, Ireland, **6** Department of Pathology, Coombe Women & Infants University Hospital, Dublin, Ireland, **7** School of Pharmacy and Medical Sciences, University of South Australia, Adelaide, Australia, **8** School of Health and Biomedical Sciences, Royal Melbourne Institute of Technology, Bundoora, Australia

‡ These authors are joint senior authors on this work.
* batesm1@tcd.ie

**Data Availability Statement:** All relevant data is within the paper and its Supporting information files. The Affymetrix microarray data sets generated as part of this study are available in the

## Abstract

Despite the use of front-line anticancer drugs such as paclitaxel for ovarian cancer treatment, mortality rates have remained almost unchanged for the past three decades and the majority of patients will develop recurrent chemoresistant disease which remains largely untreatable. Overcoming chemoresistance or preventing its onset in the first instance remains one of the major challenges for ovarian cancer research. In this study, we demonstrate a key link between senescence and inflammation and how this complex network involving the biomarkers MAD2, TLR4 and MyD88 drives paclitaxel resistance in ovarian cancer. This was investigated using siRNA knockdown of MAD2, TLR4 and MyD88 in two ovarian cancer cell lines, A2780 and SKOV-3 cells and overexpression of MyD88 in A2780 cells. Interestingly, siRNA knockdown of MAD2 led to a significant increase in TLR4 gene expression, this was coupled with the development of a highly paclitaxel-resistant cell phenotype. Additionally, siRNA knockdown of MAD2 or TLR4 in the serous ovarian cell model OVCAR-3 resulted in a significant increase in TLR4 or MAD2 expression respectively. Microarray analysis of SKOV-3 cells following knockdown of TLR4 or MAD2 highlighted a number of significantly altered biological processes including EMT, complement, coagulation, proliferation and survival, ECM remodelling, olfactory receptor signalling, ErbB signalling, DNA packaging, Insulin-like growth factor signalling, ion transport and alteration of components of the cytoskeleton. Cross comparison of the microarray data sets identified 7 overlapping genes including MMP13, ACTBL2, AMTN, PLXDC2, LYZL1, CCBE1 and CKS2. These results demonstrate an important link between these biomarkers, which to our knowledge has never before been shown in ovarian cancer. In the future, we hope that triaging patients into alterative treatment groups based on the expression of these three biomarkers or therapeutic targeting of the

ArrayExpress repository using accession number E-MTAB-8440 (direct link: https://www.ebi.ac.uk/arrayexpress/experiments/E-MTAB-8440).

**Funding:** NO - This research was supported by a research grant from the Royal City of Dublin Hospital Trust, the Emer Casey Foundation, SOCK and the Irish Ladies Golf Union. The funders had no role in study design, data collection and analysis, decision to publish, or preparation of the manuscript.

**Competing interests:** The authors have declared that no competing interests exist.

mechanisms they are involved in will lead to improvements in patient outcome and prevent the development of chemoresistance.

## Introduction

Ovarian cancer is a major cause of cancer death in women worldwide with less than 40% of women surviving beyond 5 years post-diagnosis [1]. This is due mainly to the development of recurrent chemoresistant disease which cannot as of yet be effectively treated in patients once it develops [2]. To improve patient outcomes, we must be able to either effectively destroy chemoresistant tumours once they resurface or prevent them from developing in the first instance. One way in which their development could be prevented is through the use of prognostic biomarkers which can identify patients who will likely develop chemoresistance prior to the commencement of treatment. These patients once identified could be selected out from the main patient population and given more appropriate treatments to prevent the onset of chemoresistance. In recent years our group and others have extensively investigated three new prognostic biomarkers, known as toll-like receptor 4 (TLR4), myeloid differentiation factor 88 (MyD88) and mitotic arrest deficient 2 (MAD2) for the most common and lethal form of ovarian cancer; high grade serous ovarian cancer (HGSOC). All three markers have been shown to be involved in the development of chemoresistance to paclitaxel [3–10], one of the first-line chemotherapies used to treat ovarian cancer and their expression levels have been shown to correlate with poor clinical outcome in patients [6, 7, 11–16]. TLR4 is an innate immune receptor responsible for the recognition of lipopolysaccharide (LPS) on gram-negative bacteria. Upon ligand engagement, TLR4 activates inflammatory cytokine production through its downstream adaptor molecule MyD88. Activation of this signalling pathway is thought to drive tumour-associated inflammation, resistance to apoptosis and promote the induction of a stem-like phenotype [3, 17–19]. Paclitaxel, due to its homology to LPS [20] and its ability to bind TLR4 and activate downstream signalling, is thought to promote the development of this aggressive phenotype [17, 18, 21, 22]. Elevated expression levels of TLR4 or it's adaptor protein MyD88 have been associated with reduced survival outcome in HGSOC patients [6, 8, 15, 23–25], while therapeutic targeting of TLR4 has been shown to restore paclitaxel sensitivity in ovarian cancer cell models [6, 16]. Although, it must be acknowledged that a recent largescale study found no prognostic association between TLR4 expression and HGSOC [24], despite previous contra-indications in a smaller study by the same group, particularly when TLR4 was combined with MyD88 [23]. Interestingly however the TLR4 downstream adaptor molecule MyD88 was found to be prognostic in this largescale cohort in agreement with a number of other studies including our own [6, 8, 15, 23–25]. Given these findings and the fact that paclitaxel is a known ligand for TLR4 [20], further interrogation of how this pathway contributes to paclitaxel chemoresistance is warranted.

MAD2 is a key component of the spindle assembly checkpoint (SAC) responsible for correct segregation of chromosomes during cell division. Suppression of MAD2 leads to mitotic catastrophe as cells divide without proper chromosomal segregation. This leads to anaphase bridge formation and generation of a DNA damage response which mimics normal telomere shortening resulting in the induction of cellular senescence [5, 26]. Cellular senescence allows tumour cells to resist paclitaxel which only targets actively dividing cells while also promoting tumour growth through the release of a milieu of over 40 different cytokines/chemokines and other factors as part of what is known as the senescence associated secretory phenotype

(SASP) [27]. As all three biomarkers have been shown, individually, to have a significant impact on patient prognosis and the modulation of paclitaxel chemoresponsiveness and also given the fact that many cytokines secreted during senescence are also known downstream targets of TLR4-MyD88 signalling we hypothesised that there may be crosstalk between these three important biomarkers in ovarian cancer. The aim of this study, therefore, was to assess whether there was any molecular link between MAD2, TLR4 and MyD88 in ovarian cancer and to further explore the mechanisms each of these biomarkers utilise, in order to render ovarian cancer cells resistant to paclitaxel therapy.

## Results

### Identifying the molecular link between MAD2 and TLR4-MyD88 signalling

In order to discern a possible relationship between MAD2 and TLR4-MyD88 signalling, transfection experiments were performed initially in both A2780 (MyD88 null) and SKOV-3 (MyD88 positive) ovarian cancer cells (Fig 1). Firstly, TLR4 was knocked down in both cell models using siRNA. Secondly MyD88 was knocked down in SKOV-3 cells while A2780 cells were transfected with a MyD88 overexpression plasmid. Following each transfection experiment MAD2 expression levels were assessed. Knockdown of TLR4 in both cell models did not alter MAD2 expression levels nor did knockdown or overexpression of MyD88 in SKOV-3 or A2780 cell lines respectively. Thus, indicating that TLR4-MyD88 signalling and MAD2 were independent or at the very least that MAD2 expression was not influenced by changes in TLR4 or MyD88 expression in these cell models. In parallel with this work, *in-silico* analysis was performed using the Search Tool for the Retrieval of Interacting Genes/Proteins (STRING) v10 software [28] in order to identify any potential interaction between the TLR4-MyD88 signalling pathway and MAD2. In support of the transfection experiments *in-silico* analysis identified no direct relationship between MAD2 and TLR4 or MyD88. TLR4, MyD88 and their interactants segregated into entirely different clusters than MAD2 and its interactants (Fig 1C).

These initial transfection experiments and the *in-silico* analysis supported the idea that TLR4-MyD88 signalling and MAD2 acted as independent biomarkers in ovarian cancer. However, to conclusively demonstrate this, in the reverse setting, TLR4 and MyD88 expression was analysed following knockdown of MAD2 in both A2780 and SKOV-3 cells (Fig 2). Most interestingly when MAD2 levels were suppressed using siRNA, both A2780 and SKOV-3 cells exhibited a significant 3-fold increase in TLR4 gene expression, demonstrating a previously never-before shown link between TLR4 and MAD2 in ovarian cancer (Fig 2A and 2D). However, surprisingly a similar increase in TLR4 protein expression post knockdown of MAD2 was not observed at the selected timepoint in either cell line (Fig 2B and 2E).

To further explore the link between TLR4 and MAD2 we next analysed the expression of MAD2, TLR4 and MyD88 in 5 additional ovarian cancer cell lines; OVCAR-3, PEO1, OAW42, KURAMOCHI and 59M cells (Fig 3A). Of these only OVCAR-3 and PEO1 expressed TLR4, MyD88 and MAD2. OAW42, KURAMOCHI and 59M were TLR4 negative. OVCAR-3 cells due to their TLR4 positivity and as a representative model of serous ovarian cancer were subsequently transfected with siRNA targeting TLR4 or MAD2 and then TLR4, MAD2 and MyD88 expression levels were assessed. Interestingly knockdown of TLR4 or MAD2 in the OVCAR-3 cell model caused a significant 2.4 and 2.9 fold increase in MAD2 or TLR4 expression respectively further highlighting an important link between these two biomarkers (Fig 3B).

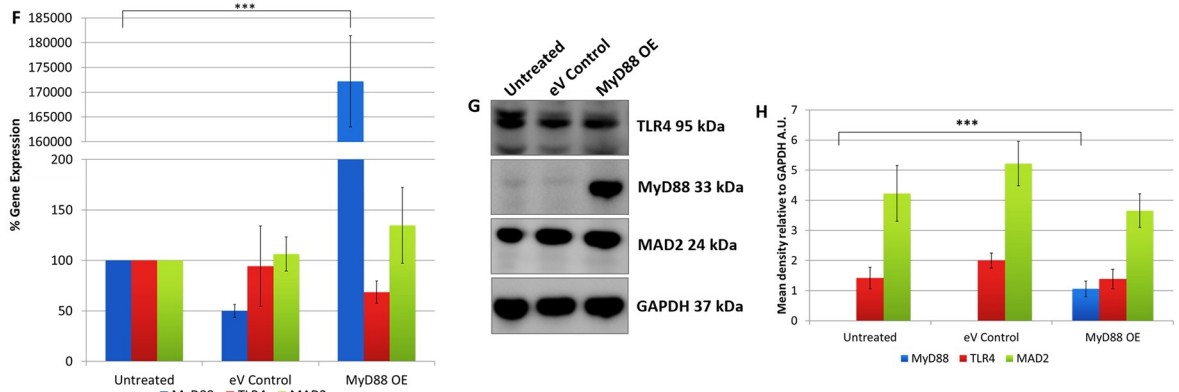

**Fig 1. Alteration of TLR4 and MyD88 expression does not alter MAD2 expression in A2780 or SKOV-3 cells.** MyD88, TLR4 and MAD2 gene expression levels in (A) A2780 and (B) SKOV-3 cells, 72 hours following transfection with siRNA targeting TLR4 or MyD88. Neither knockdown of TLR4 in both SKOV-3 and A2780 cells or knockdown of MyD88 in SKOV-3 cells had any significant impact on the expression of MAD2. (C) Screenshots from the STRING website, which was queried for relationships between TLR4, MyD88 and MAD2. Coloured lines between the proteins indicate the various types of interaction evidence. *In-silico* analysis predicted that there was no direct interaction between MAD2 and MyD88 or TLR4. (D) Western blot analysis and (E) densitometric analysis of MAD2 protein expression

levels in SKOV-3 cells following transfection with siRNA targeting MyD88 or TLR4. (F) MyD88, TLR4 and MAD2 gene expression levels, (G) western blot analysis and (H) densitometric analysis of A2780 cells transfected with a MyD88 overexpression plasmid for 72 hours. The results demonstrate that overexpression of MyD88 had no significant impact on MAD2 gene or protein expression. Results are expressed as mean +/-SD, at least n = 3; NS—Not significant, *p<0.05, **p<0.01, ***p<0.01 (Student's t-test). Densitometry results are expressed in arbitrary units (A.U) normalised to GAPDH. **Note**:- Blots are cropped from original images available in the S1 Raw Images.

## Suppression of MAD2 induces cellular senescence and paclitaxel resistance

Following knockdown of MAD2, SKOV-3 cells exhibited an increase in cell and nuclear size and alteration of cell shape (Fig 4A). Subsequently, the chemoresponse of SKOV-3 cells to paclitaxel following knockdown of MAD2 was assessed (Fig 4C). When SKOV-3 cells were treated with a 20nM or 1μM dose of paclitaxel following knockdown of MAD2, they exhibited a reduction in cell viability of 36.2% and 36.1% compared with untransfected cells which were not treated with paclitaxel. In contrast, untransfected cells or cells transfected with the scrambled negative control which were treated with 20nM of paclitaxel exhibited a decrease in cell viability of 55.4% and 56.3% respectively. While untransfected cells or cells transfected with the scrambled negative control siRNA which were treated with 1μM of paclitaxel exhibited a decrease in cell viability of 66.2% and 66.4% respectively. Additionally, despite there being no

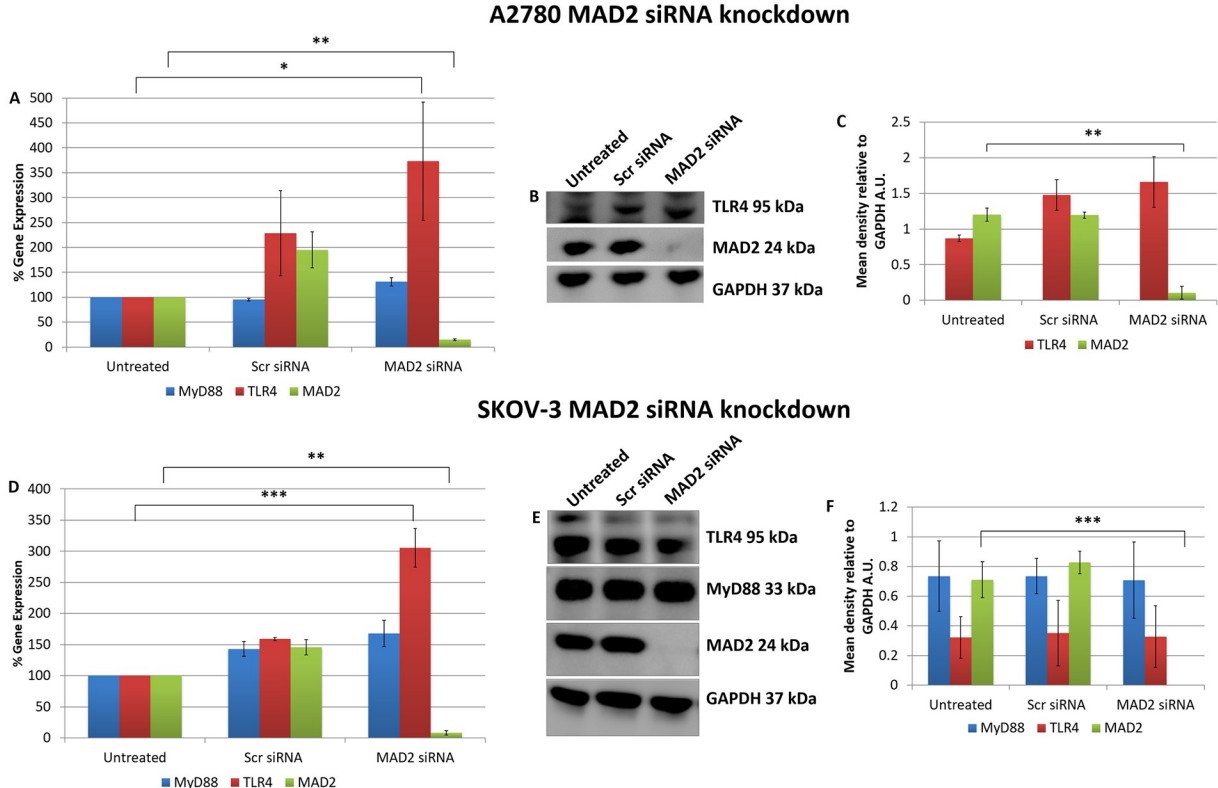

**Fig 2. MAD2 is an inhibitor of TLR4 gene expression.** MyD88, TLR4 and MAD2 gene expression levels in (A) A2780 and (D) SKOV-3 cells, 72 hours following transfection with siRNA targeting MAD2. Interestingly TLR4 gene expression, but not that of its adaptor molecule MyD88, was increased 3-fold following siRNA knockdown of MAD2 for 72 hours in both cell lines. (B) Western blot and (C) densitometry analysis of protein lysates harvested from A2780 cells following knockdown of MAD2 revealed however that suppression of MAD2 had no impact on TLR4 or MyD88 protein expression. Similarly, western blot (E) and densitometric analysis (F) in SKOV-3 cells found no significant increase in TLR4 protein expression post knockdown of MAD2. Results are expressed as mean +/-SD, n = 3; *p<0.05, **p<0.01, ***p<0.001 (Student's t-test). Densitometry results are expressed in arbitrary units (A.U) normalised to GAPDH. **Note**:- Blots are cropped from original images available in the S1 Raw Images.

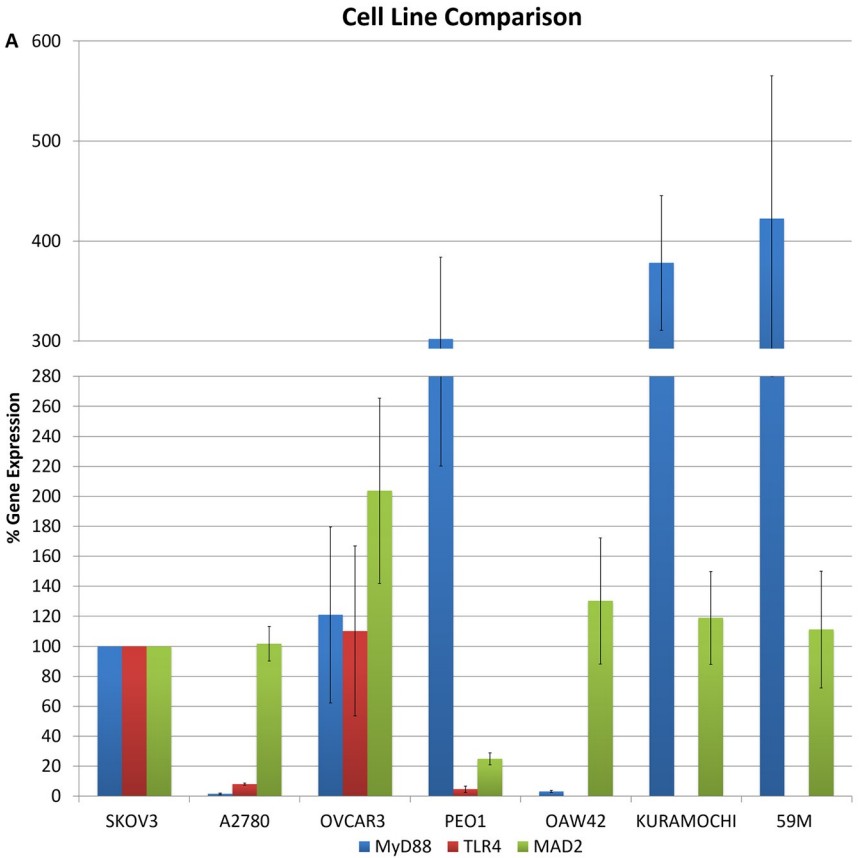

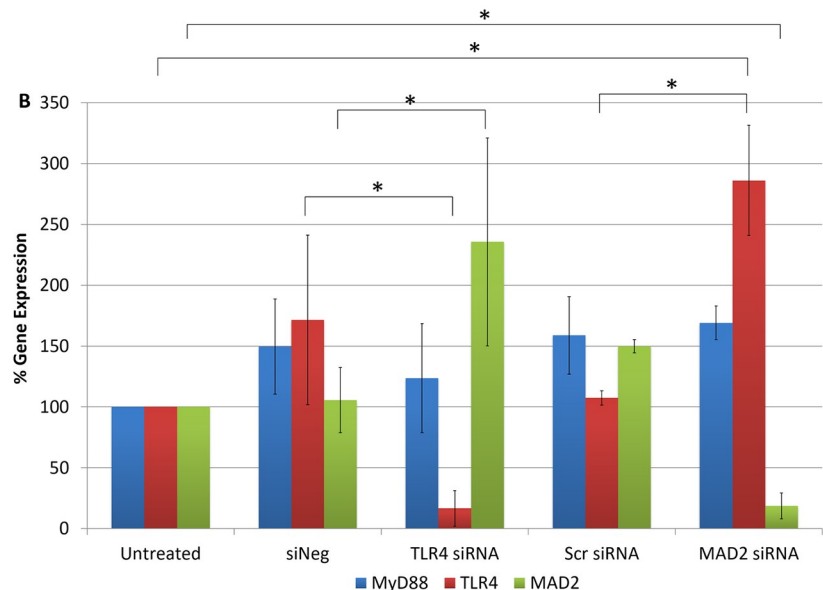

**Fig 3. Cell line cross comparison and OVCAR-3 TLR4 and MAD2 siRNA knockdown.** (A) MyD88, TLR4 and MAD2 gene expression in A2780, OVCAR-3, PEO1, OAW42, KURAMOCHI and 59M cells relative to SKOV-3 cells. (B) MyD88, TLR4 and MAD2 gene expression in OVCAR-3 cells following siRNA knockdown of TLR4 or MAD2. Gene expression levels were normalised to the endogenous control GAPDH and calibrated to that of untreated cells to establish the relative change in gene expression.

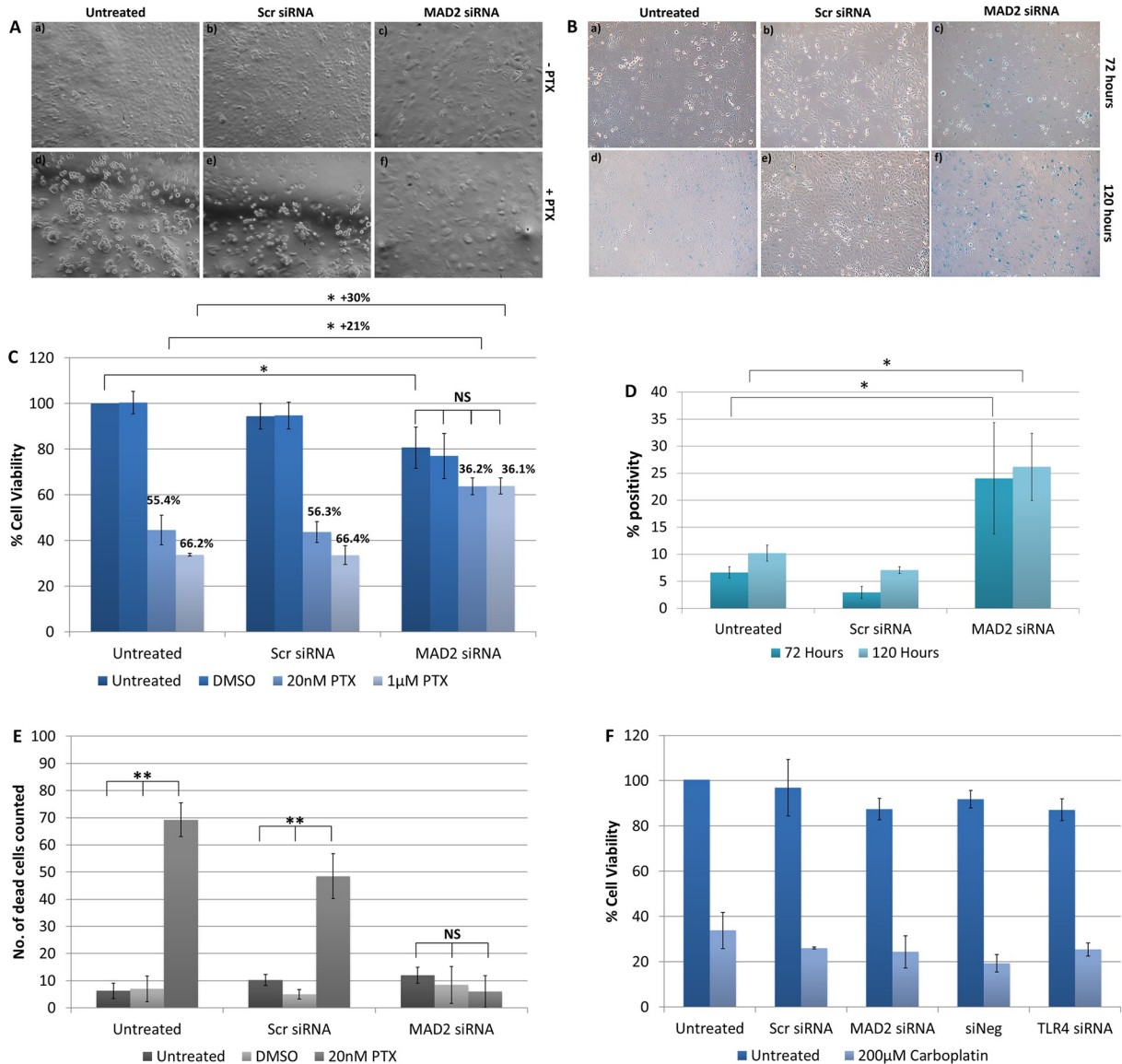

**Fig 4. Suppression of MAD2 induces cellular senescence and paclitaxel resistance in SKOV-3 cells.** (A) Representative images of untransfected SKOV-3 cells and SKOV-3 cells transfected with siRNA targeting MAD2 (MAD2 siRNA) or a scrambled negative control siRNA (Scr siRNA) for 72 hours. After 24 hours, cells were either left untreated (-PTX) (a-c) or were treated with a 1µM dose of paclitaxel (+PTX) (d-f) for a further 48 hours. (B) Representative images of SKOV-3 cells stained using the senescence β-galactosidase staining kit following transfection for 72 hours (a-c) or 120 hours (d-f). (C) CCK-8 assay results. % cell viability for each condition was calculated as a percentage of untransfected SKOV-3 cells which were left untreated. The results demonstrate that knockdown of MAD2 renders SKOV-3 cells resistant to paclitaxel. (D) The percentage of β-galactosidase positive cells was calculated for each condition for (n = 3) technical and (n = 3) biological replicates. Following transfection a 3-fold increase in β-galactosidase expression was observed demonstrating that knockdown of MAD2 induces cellular senescence in SKOV-3 cells. (E) Trypan blue exclusion assay. (F) CCK-8 assay results for SKOV-3 cells treated with 200µM carboplatin for 48 hours following a 24 hour transfection. Results are expressed as mean +/-SD, n = 3. *p<0.05, **p<0.01, ***p<0.001 (Student's t-test).

visual signs of cytotoxicity, untreated cells transfected with MAD2 siRNA also exhibited a 19% significant reduction in cell viability compared to untransfected cells which were untreated potentially indicating a reduction in cell proliferation. Furthermore, in transfected cells treated with either dose of paclitaxel minimal if any visual signs of cytotoxicity were observed. In fact, the addition of paclitaxel appeared to accelerate the timeframe for the emergence of the

enlarged cell phenotype. Additionally, the difference in cell viability between cells transfected with MAD2 siRNA which were untreated and transfected cells treated with either dose of paclitaxel was not statistically significant. This result was further supported by a trypan blue exclusion assay which detected a significant decrease in the number of dead cells in the supernatants of paclitaxel treated cells following knockdown of MAD2 compared to controls (Fig 4E). The results indicated that SKOV-3 cells transfected with siRNA targeting MAD2 were rendered resistant to paclitaxel and potentially undergoing cellular senescence. In order to investigate this further, SKOV-3 cells were stained with the senescence β-galactosidase staining kit. The number of cells which were β-galactosidase positive were counted and then compared against background levels in negative control and untreated cells (Fig 4B). A three-fold increase in the percentage of β-galactosidase positive cells was observed following knockdown of MAD2 compared to the untransfected and scramble negative controls (p<0.001), which was sufficient to indicate the induction of cellular senescence (Fig 4D). We also assessed the cytotoxicity of SKOV-3 cells to carboplatin following knockdown of either TLR4 or MAD2. Cells were transfected for 24 hours and then treated with a 200μM dose of carboplatin or left untreated, however, neither knockdown of TLR4 or MAD2 altered the response of SKOV-3 cells to carboplatin (Fig 4F).

Following these interesting results, High Mobility Group Box 1 (HMGB1) gene and protein expression was assessed post knockdown of MAD2 (Fig 5). HMGB1 was investigated as it had previously been shown to directly upregulate TLR4 expression [29] and due to the fact that its secretion during the early stages of senescence is known to be key to the formation of a SASP [30, 31]. However, no difference in HMGB1 expression was detected in either cell line and

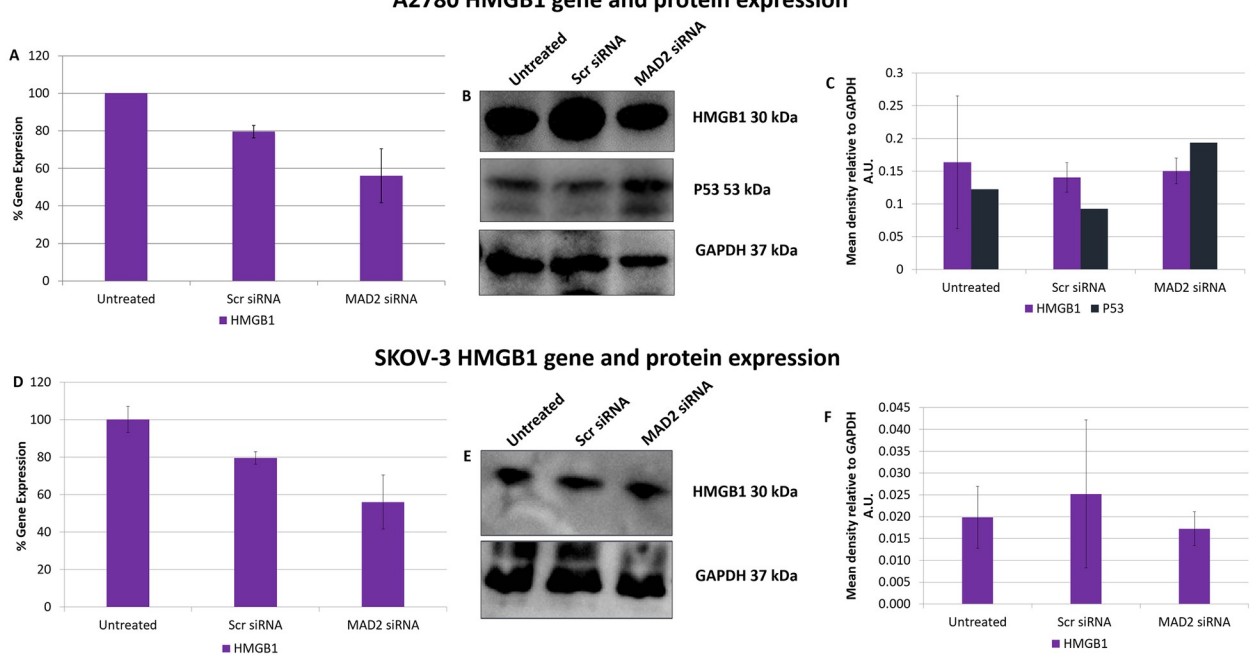

**Fig 5. SiRNA knockdown of MAD2 does not alter HMGB1 gene or protein expression in A2780 or SKOV-3 cells.** HMGB1 gene expression levels in A2780 (A) and SKOV-3 cells (D) following a 72 hour knockdown of MAD2. HMGB1 and P53 protein expression was also examined in A2780 cells using western blot analysis (B) and densitometry (C) while HMGB1 protein expression only was examined in P53 null SKOV-3 cells using western blot analysis (E) and densitometry (F). No significant change in HMGB1 gene or protein expression was observed in SKOV-3 cells nor was any change observed in HMGB1 gene expression or HMGB1 or P53 protein expression in A2780 cells post knockdown of MAD2 for 72 hours. Results are expressed as mean +/-SD. **Note**:- Blots are cropped from original images available in the S1 Raw Images.

further evidence needs to be gathered to support the hypothesis that HMGB1 acts as the link between MAD2 mediated senescence and TLR4 upregulation.

## Microarray analysis

As the knockdown of TLR4 had been shown to enhance the sensitivity of SKOV-3 cells to paclitaxel and the knockdown of MAD2 had been shown to render these cells paclitaxel resistant it was decided to perform microarray analysis post knockdown of TLR4 or MAD2 in this cell model. This was done in order to further discern any links between these biomarkers and gain greater insight into how they modulate the cellular response to paclitaxel. Following knockdown of TLR4 a total of 166 protein coding targets were found to be significantly upregulated and 286 targets found to be significantly (S1 Data). The differentially expressed genes identified following knockdown of TLR4 were subsequently analysed using the online gene ontology database DAVID, in order to identify important biological processes in which these genes participate [32]. A number of important biological processes were highlighted including cell death, cell adhesion, steroid biosynthesis and metabolism, complement and coagulation cascades and ErbB signalling among others (Fig 6 and S1 Table).

Following knockdown of MAD2,126 protein coding genes were found to be upregulated and 95 protein coding genes were found to be downregulated (S2 Data). Microarray analysis highlighted several features of senescence, which were deregulated following knockdown of MAD2. These included an effect on DNA packaging, lipase activity, ion transporter activity, Insulin-like growth factor binding protein (IGFBP) activity, arachidonic acid metabolism, regulation of cell motility and migration, ossification and bone metabolism, the sensory perception of smell and the response to hormones and various chemical and extracellular stimuli (S2 Table). The complete microarray data sets for both knockdown experiments are available at ArrayExpress (Accession #370077). Cross-comparison of the microarray data sets highlighted 12 common genes which were deregulated in both data sets. This can be observed along with a map of the entire network of differentially expressed genes highlighted in both data sets in (Fig 7).

## Discussion

Our group and others have previously demonstrated that TLR4, MyD88 and MAD2 play key roles in paclitaxel resistance in ovarian cancer and are associated with poor patient outcome [3, 4, 6–9, 11–13, 15, 16, 24, 25, 33–36]. Although the recent and impressively sized study by Block et al. [24] demonstrated that TLR4 was not prognostic, several other studies including our own have previously demonstrated that TLR4 is linked to poor patient outcome and this trend can even be observed in the wider cancer space, with similar patterns observed in breast [10, 37], oesophageal [38] and other cancer types [39]. Block and colleagues [24] did find however that the TLR4 downstream adaptor protein MyD88 was prognostic, which is the signalling arm of the TLR4 pathway used to modulate the response of SKOV-3 cells to paclitaxel. This study sought to further explore how the TLR4-MyD88 signalling pathway and MAD2 mediated senescence contribute to the cellular response to paclitaxel and discern any molecular link between these three biomarkers as well as identifying new markers for potential future therapeutic exploitation. TLR4 itself is potentially targetable, a previous *in-vitro* study found that targeting TLR4-MyD88 signalling using the small molecule Atractylenolide-I could resensitise cells to paclitaxel [16]. Other TLR4 antagonists such as TAK-242 and Eritoran have also been examined in clinical trials for sepsis/inflammation as reviewed in [40] and may be suitable for ovarian cancer patients overexpressing TLR4 or MyD88. MyD88 is also targetable and such treatments may potentially eradicate paclitaxel resistant cancer stem cells (CSCs) by

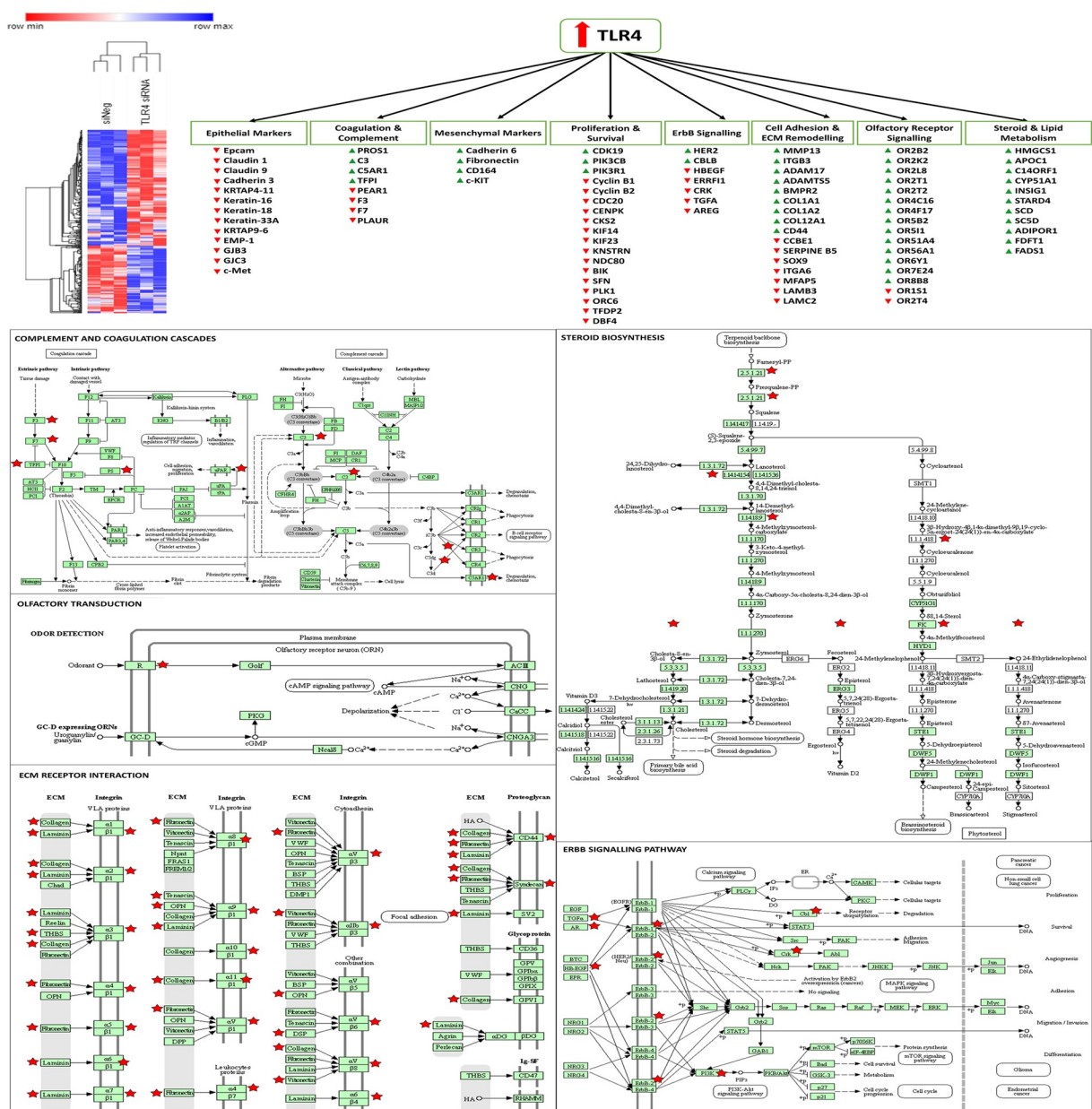

**Fig 6. Altered genes and biological processes following knockdown of TLR4.** Microarray analysis revealed that TLR4 controls genes related to EMT, survival and proliferation, steroid and lipid metabolism, olfactory receptor signalling, adhesion, coagulation and complement cascades, and ErbB signalling. A hierarchial clustering heatmap was generated using Morpheus from the broad institute. KEGG pathway maps were generated using DAVID, red stars indicate genes significantly altered in the TLR4 knockdown microarray data set.

inducing differentiation [41, 42]. A wide array of inhibitors are also available for the TLR4-MyD88 pathway downstream transcription factor NFκB [43]. In this study suppression of TLR4 or alteration of MyD88 expression in either SKOV-3 or A2780 cells had no resulting impact on MAD2 expression. Intererestingly however in the OVCAR-3 serous ovarian model a significant increase in MAD2 expression was observed. One observation with this experiment was that the replicate with the highest knockdown of TLR4 had the highest upregulation of MAD2. Furthermore with both the SKOV-3 and A2780 cells lines MAD2 expression was

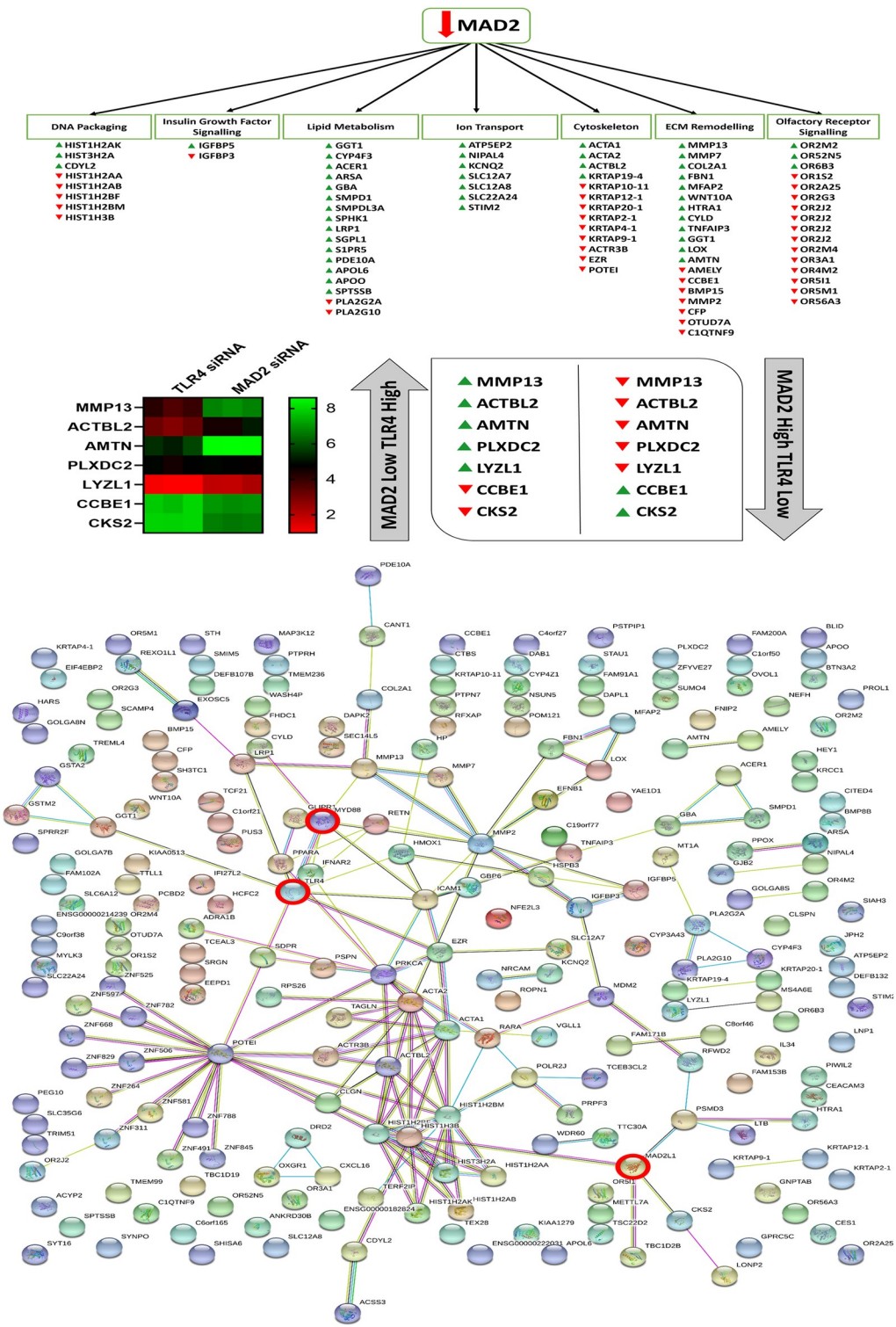

**Fig 7. Altered genes and biological processes following knockdown of MAD2, string network analysis and cross comparison of the MAD2 and TLR4 knockdown data sets.** Microarray analysis revealed that suppression of MAD2 and the induction of senescence affects various genes involved in DNA packaging, lipid metabolism, ion transport, Insulin-like growth factor signalling, ECM remodelling and olfactory receptor signalling and components of the actin cytoskeleton. Cross comparison of the two microarray data sets identified 7 key genes which are differentially regulated in both data sets. String network analysis plots of each of the altered genes identified in both data sets and how they relate to each other and TLR4, MyD88 and MAD2 which are highlighted by red circles.

partially increased ~1.5 fold post knockdown of TLR4, however this was not statistically significant. The observed differences may indicate that TLR4 needs to be supressed below a certain threshold to have a substantial impact on MAD2 expression. Increasing the efficacy of the TLR4 knockdown, through the use of shRNA vectors and cell selection or the use of newer technologies such as CRISPR may yield more definitive evidence for this.

Furthermore, suppression of MAD2 expression using siRNA in A2780, OVCAR-3 and SKOV-3 cell lines led to a significant increase in TLR4 gene expression levels demonstrating a key link between TLR4 and MAD2. Although it must be acknowledged that although TLR4 gene expression was upregulated post knockdown of MAD2, there was no recipient increase at the protein level in SKOV-3 or A2780 cells. This is a curious result, although TLR4 protein expression following knockdown of MAD2 was only examined at a single timepoint, 72 hours post transfection in both cell lines. However, there are two potential biological mechanisms which could explain why the recipient increase in TLR4 expression at the protein level was not observed. Firstly, the activation of the unfolded protein response as is known to occur during cellular senescence may have blocked or limited the amount of protein translation taking place [44]. Secondly, TLR4 shedding can occur as a result of oxidative stress [45] and during cellular senescence reactive oxygen species (ROS) levels are known to be dramatically increased [27]. Indeed, TLR4 shedding may even represent a feedback mechanism to blunt hyper-reactive TLR4-ligand signalling which may occur due to HMGB1 release which also occurs during senescence [31, 46]. HMGB1 was examined as part of this study however its expression was not found to be altered post knockdown, which may be as a result of its function as a secreted cytokine. TLR4 is also known to be upregulated by various other cytokines which may be secreted as part of the SASP [47–49].

A number of interesting target genes were highlighted during the microarray analysis in SKOV-3 cells following knockdown of TLR4 associated with metastasis, angiogenesis, EMT/differentiation and circulating tumour cell (CTC) biology including CD44, HER2, PI3K, MMP13, members of the claudin, cadherin, integrin and laminin family and various olfactory receptors (ORs). Although these markers require further validation, many of them have previously been shown to be prognostic for ovarian and other types of cancer and are potentially targetable [41, 42, 50–54]. TLR4 also appears to be suppressing both coagulation and complement which again may contribute to metastasis and therapeutic resistance [55–62]. Targeting of another marker identified on the arrays in the steroid biosynthesis group, farnesyl transferase has also previously been shown to enhance paclitaxel sensitivity in ovarian models [63] and has previously been targeted in clinical trials [64]. Some of the other steroid pathway targets have been shown to control vesicular trafficking of HER2 and related family members [65]. The microarray analysis indicates that the TLR4-MyD88 signalling pathway is likely driving paclitaxel resistance by promoting the induction of an EMT/stem like phenotype. Ligation of TLR4 by paclitaxel also likely provides a further survival advantage to these cells by upregulating pro-survival signalling molecules including pAKT, BCL-2, BCL-XL and XIAP as reported previously [7, 8, 37]. Given the crosstalk identified between MAD2 and TLR4 in this study it is likely that senescence is also helping to drive/amplify this phenomenon. Clearly new therapeutic strategies which actively target these mechanisms need to be introduced in order to enhance the efficacy of current ovarian cancer therapy.

Knockdown of MAD2 in the SKOV-3 cell model revealed a number of altered senescence associated genes and processes [27, 66–72]. Among those affected were genes involved in OR activity and the response to a number of different chemical and extracellular stimuli, IGFBP activity and ossification, cell motility, lipase & phospholipase activity and arachidonic acid metabolism, DNA packaging and ion transporter activity. Senescent cells likely act as a protective barrier against paclitaxel and perhaps other chemotherapeutic agents, shielding

non-senescent populations of cancer cells such as CSCs from drug-induced cell death while simultaneously promoting their growth through the milieu of cytokines they release as part of the SASP [26, 27]. Previous reports have demonstrated that senescent cells are capable of promoting the growth of tumours and induce the progression of pre-malignant lesions into malignant tumours in *in-vivo* xenografts [73, 74]. They may also promote tumour growth through active suppression of immune cell populations [75]. Thus, senescence and an activated TLR4 signalling pathway likely promotes tumour through the generation of inflammatory niche which selects for invasive paclitaxel resistant CSC populations leading to shorter survival time in patients. Cross-comparison of microarray data sets highlighted several genes involved in cell adhesion, proliferation, differentiation, migration and extracellular matrix (ECM) degradation [76–86]. Targeting some of the markers highlighted in the arrays or treating patients with exogenous MAD2 may help to reverse the senescence phenotype and restore paclitaxel sensitivity. We also previously identified that the MAD2 regulatory microRNA miR-433 was dysregulated and associated with poor prognosis in ovarian cancer patients and may act as an upstream inducer of MAD2 mediated senescence [11]. Therefore, these patients may benefit from miR-433 antagomir therapy. A variety of other anti-senescence therapies are available which may help boost paclitaxel efficacy when MAD2 or miR-433 are used as triage markers [26, 87]. The results also demonstrate that these senescent cell populations appear to be selectively resistant to paclitaxel but are sensitive to carboplatin, therefore patients could potentially be selected out for single arm therapy with carboplatin.

## Conclusions

The molecular link between TLR4-MyD88 signalling and MAD2 identified in this study has potentially important implications for the development of new treatment strategies for ovarian cancer patients. Individually these markers highlight paclitaxel resistance mechanisms within a patient's tumour. Depending on the expression of these markers, one or multiple mechanisms may need to be targeted. The complexity of downstream signalling pathways identified by microarray analysis also further highlight the fact that a single biomarker alone may be insufficient to capture the multiple pathophysiological processes occurring within a patient's tumour which contribute to chemoresistance. Assessing multiple biomarkers such as TLR4, MyD88 and MAD2 which give greater insight into the pathological makeup of a patient's tumour may help to direct therapies and more suitable treatment combinations in order to improve overall outcome.

## Methods

### Cell culture

A2780, OAW42, OVCAR-3, PEO1 and KURAMOCHI cells were cultured in RPMI 1640 medium (Sigma Aldrich, St Louis, USA), 59M cells were cultured in DMEM (Sigma Aldrich) and SKOV-3 cells were cultured in McCoys modified 5A medium (Sigma Aldrich) respectively. All media was supplemented with 10% foetal bovine serum (FBS) (Sigma Aldrich) and 2% penicillin/streptomycin (5000IU, Sigma Aldrich) and cells were maintained in a humidified atmosphere at 37°C and 5% $CO_2$.

### Small interfering RNA transfection

siRNA targeting TLR4 (TLR4 siRNA, s14194), MyD88 (MyD88 siRNA, s9136) and silencer select negative control #1 siRNA (siNeg, 4390843) were purchased from (Thermo Fisher Scientific, Waltham, USA) and on target plus SMARTpool MAD2L1 siRNA (MAD2 siRNA,

L-003271-00-0005) and on target plus SMARTpool non-targeting siRNA (Scr siRNA, D-001810-01-05) were purchased from (Dharmacon, Lafayette, USA). SKOV-3 cells were transfected into either 24 or 6 well plates at seeding densities of 25,000 or 125,000 cells per well respectively. Cells were transfected with Lipofectamine RNAiMAX (13778–075, Thermo Fisher Scientific), Opti-MEM® I reduced serum medium (31985–047, Thermo Fisher Scientific) and siRNA at a final concentration of 1nM per well. A2780 cells were transfected into 6 well plates at a seeding density of 400,000 cells per well and were transfected with siRNA targeting TLR4 or MAD2 at final concentrations of 10nM or 30nM per well respectively. OVCAR-3 cells were transfected into 24 well plates at a seeding density of 25,000 cells per well. OVCAR-3 cells were transfected with 30nM of siRNA targeting TLR4 or MAD2. All transfections were carried out using media not containing antibiotics.

## MyD88 transfections

For the MyD88 transfection experiments A2780 cells were transfected with a MyD88 overexpression plasmid (MyD88 OE) or an empty vector negative control plasmid (eV Control), both purchased from IMAgenes, or were left untreated for 72 hours. For each transfection experiment, plasmid DNA and lipofectamine was first diluted in Opti-MEM® I reduced serum medium. A2780 cells were transfected into 6 well plates at a seeding density of 400,000 cells per well. All transfections were carried out using media not containing antibiotics. The final plasmid DNA concentration per well was 1ng/ul.

## RNA extraction and TaqMan RT-PCR

Total RNA was isolated as per the manufacturer's instructions using the *mir*Vana™ miRNA Isolation Kit (AM1560, Thermo Fisher Scientific). RNA concentration was determined using a nanodrop 2000c spectrophotometer (Thermo Fisher Scientific). Reverse transcription was carried out using the High Capacity cDNA Reverse Transcription Kit (4368813, Thermo Fisher Scientific) on the Gene Amp PCR System 9600 (Perkin Elmer, Waltham, USA). TaqMan RT-PCR was then performed using the 7900HT Real-Time PCR System (Thermo Fisher Scientific). Primers and probes for TLR4 (Hs00152939_m1), MAD2 (H203063324_g1), MyD88 (Hs00182082_m1), HMGB1 (Hs01037385_s1) and the endogenous controls, glyceraldehyde 3-phosphate dehydrogenase (GAPDH, 4333764T) or Beta-2 Microglobulin (B2M, 4333766F) were obtained from (Thermo Fisher Scientific). These are supplied as commercial predesigned primer and probe mixes (20X). Gene expression levels following transfection were calculated using the ΔΔCT method relative to the endogenous control [88]. A significant change in gene expression was considered to be present if at least a 2-fold change (above 200% expression or below 50% expression) in gene expression was observed, with a p value of ≤0.05 compared to untreated cells and/or negative control cells.

## Western blot analysis

Following transfection for 72 hours protein was extracted from SKOV-3 cells using RIPA lysis buffer (Sc-24948, Santa Cruz Biotechnology, Santa Cruz, USA) modified with phenylmethanesulfonyl fluoride (PMSF) (200mM), a protease inhibitor cocktail, and sodium orthovanadate ($Na_3VO_4$) (100mM). Cell suspensions were later sonicated to ensure complete lysis using the soniprep 150 (MSE Labs, East Sussex, UK). Protein concentration was then determined using the Pierce™ BCA Protein Assay Kit (23225, Thermo Fisher Scientific). 30μg of protein samples were then resolved by SDS-PAGE on 4–12% Bis-Tris NuPage gels (NP0321, Thermo Fisher Scientific) using the XCell SureLock® Mini-Cell SDS PAGE rig (Thermo Fisher Scientific).

Resolved proteins were then transferred to 0.2μM Hybond PVDF membranes (10600021, Amersham, Amersham, UK) using the XCell II™ Blot Module (Thermo Fisher Scientific). Following transfer membranes were blocked using 5% w/v milk protein and probed using antibodies directed against TLR4 (1:100, Ab47093, Abcam), MAD2 (1: 1000, 610679, BD Biosciences), MyD88 (1:1000, D80F5, Cell Signalling Technology), P53 (1:500, sc-126, Santa Cruz Biotechnology), HMGB1 (1:50, sc-56698, Santa Cruz Biotechnology) or GAPDH (1: 10,000, Ab9485, Abcam). After washing, the membrane was incubated with either a horseradish peroxidase (HRP) linked anti-rabbit secondary antibody (#7074, 1:1000, Cell Signalling Technology) or an anti-mouse HRP-linked secondary antibody (#7076, 1:1000, Cell Signalling Technology). Following incubation with the primary and secondary antibodies, a detection reagent luminol (SC-2048, Santa Cruz Biotechnology) was applied to blots and chemiluminescence images were then developed using a LAS-4000 luminescent image analyser (Fujifilm, Minato, Japan). Molecular weight was confirmed using a MagicMark™ XP Western Protein Standard (LC5602, Thermo Fisher Scientific) and SeeBlue™ Plus2 Pre-stained Protein Standard (LC5925, Thermo Fisher Scientific). Restore™ PLUS Western Blot Stripping Buffer (46430, Thermo Fisher Scientific) was used to remove bound primary and secondary antibodies from membranes so they could be reprobed with additional antibodies. Densitometry was then carried out using Quantity One software (Bio-Rad Laboratories, Hercules, USA). Abundance of protein in arbitrary units (A.U.) was normalised to GAPDH. The mean density ratio of triplicate bands for each condition was then determined.

## Senescence β-galactosidase staining kit

The induction of senescence in cells is usually accompanied by an increase in β-galactosidase activity [89]. In order to demonstrate this, cells were stained with the senescence β-galactosidase staining kit (#9860, Cell Signalling Technology) following transfection for 72 hours and 120 hours. Images were then taken at 10X magnification using an Olympus CKX41 microscope and an Olympus E600 camera (Olympus, Shinjuku, Japan). The percentage of β-galactosidase positive cells within each image was then calculated for each condition for (n = 3) technical and (n = 3) biological replicates.

## Drug treatment and assessment of cell viability

Carboplatin (C2538), Paclitaxel (T402) and DMSO (D2650) were purchased from Sigma Aldrich. Carboplatin was diluted in sterile nuclease free water to a concentration of 10mg/ml and stored at room temperature based on recommendations by the manufacturer. Paclitaxel was diluted in DMSO to a concentration of 50g/l (58.6mM) based on recommendations by the manufacturer aliquoted and stored at -20˚C while DMSO was kept at room temperature. Aliquots of Carboplatin, Paclitaxel and DMSO were freshly diluted with media for each experiment to the desired working concentrations. Following transfection for 24 hours, SKOV-3 cells were either left untreated, treated with 0.0017% DMSO (vehicle control) or 20nM or 1μM of paclitaxel or 200μM of Carboplatin for 48 hours. Forty-eight hours post-treatment, cell viability was assessed using the cell cycle kit 8 (CCK-8) assay. Absorbance values were read at 450nm using the Sunrise™ microplate reader (Tecan Trading AG, Männedorf, Switzerland). Cell viability for each condition/drug treatment was calculated as a % of non-transfected cells which were left untreated. Images were taken at 4X magnification using an axiovert 35 inverted microscope (Zeiss, Germany) and at 6X zoom using a Canon Powershot A620 digital camera.

## Trypan blue dye exclusion assay

SKOV-3 cells were transfected into 6 well plates with siRNA targeting MAD2 a nontargeting scrambled negative control siRNA or were left untreated for 72 hours. After 72 hours, cells were left untreated, treated with DMSO or were treated with 20nM paclitaxel and incubated for a further 48 hours. After the 48-hour drug incubation time, supernatants from each well were collected and wells rinsed with PBS to remove any residual dying cells. Collected supernatant and washings were pelleted by centrifugation and resuspended in a small volume of PBS. Cell suspensions were then mixed at 1:1 ratio with trypan blue (T8154, Sigma Aldrich) and the number of dead cells were counted using a haemocytometer.

## Microarray analysis

Prior to analysing RNA samples using Affymetrix microarrays, the quality of RNA samples was assessed using the Agilent 2100 Bioanalyzer. Samples were run on chips from the RNA 6000 Nano kit (5067–1511, Agilent Technologies, Santa Clara, USA) and an RNA Integrity Number (RIN) was obtained. 250ng of each RNA sample was then converted into sense strand cDNA using the GeneChip® WT PLUS Reagent Kit (902280, Affymetrix). Each cDNA sample was then hybridised to Affymetrix GeneChip® human gene 2.0 ST arrays (902113, Affymetrix). Arrays were washed using the Affymetrix GeneChip® fluidics station 450 and scanned using the Affymetrix GeneChip® Scanner 3000. Gene array data was analysed using Bioconductor software libraries available at (www.bioconductor.org) [90] and the RMA method [91]. Differential expression analysis across all the arrays was carried out using RankProd [92]. DAVID, a free bioinformatics resource was used to characterise differentially expressed genes in order to identify molecular function and biological process-related genes through gene ontology [32]. Microarray analysis was performed using three biological replicates of SKOV-3 cells transfected with either the scrambled or negative control siRNA or siRNA targeting TLR4 or MAD2. A 1.5 fold change in gene expression and a p value of <0.05 was set as the threshold for a significantly upregulated/downregulated gene, this threshold is in line with other published works [93]. Heatmaps were generated using Morpheus (https://software.broadinstitute.org/morpheus) or Graphpad Prism v8.4. KEGG pathway maps were generated using DAVID. The Affymetrix microarray data sets generated as part of this study are available in an ArrayExpress repository, accession #370077.

## *In-silico* analysis

*In-silico* analysis was performed in order to identify any potential interaction between the TLR4-MyD88 pathway and MAD2 using the Search Tool for the Retrieval of Interacting Genes/Proteins (STRING) v10 software which is freely available at (http://string-db.org/). This free online bioinformatics resource identifies protein-protein interactions through both direct (physical) as well as indirect (functional) associations [28].

## Statistical analysis

A student's t-test was performed on all qPCR, densitometry and cell viability data to assess the statistical significance of gene silencing experiments and differences in cell viability between drug-treated versus untreated and vehicle control groups. A statistically significant difference was considered to be present at p≤0.05. Statistical analysis was performed using Microsoft Excel 2016.

## Supporting information

**S1 Data. List of differentially expressed genes following siRNA knockdown of TLR4 in SKOV-3 cells.**
(XLSX)

**S2 Data. List of differentially expressed genes following siRNA knockdown of MAD2 in SKOV-3 cells.**
(XLSX)

**S1 Raw Images. Uncropped western blots from A2780 and SKOV-3 cells.**
(PDF)

**S1 File. IC50 data for A2780 and SKOV-3 cells.**
(DOCX)

**S1 Table. Significantly over-represented biological processes identified by the DAVID database following knockdown of TLR4 in SKOV-3 cells.**
(DOCX)

**S2 Table. Features of senescence highlighted by microarray analysis following knockdown of MAD2 in SKOV-3 cells for 72 hours.**
(DOCX)

## Author Contributions

**Conceptualization:** Mark Bates, Cathy D. Spillane, Michael F. Gallagher, Amanda McCann, Cara Martin, Sharon O'Toole, John J. O'Leary.

**Data curation:** Mark Bates, Cathy D. Spillane, Michael F. Gallagher, Gordon Blackshields, Sharon O'Toole.

**Formal analysis:** Mark Bates, Cathy D. Spillane, Michael F. Gallagher, Gordon Blackshields, Helen Keegan, Sharon O'Toole.

**Funding acquisition:** Cathy D. Spillane, Michael F. Gallagher, Sharon O'Toole, John J. O'Leary.

**Investigation:** Mark Bates.

**Methodology:** Mark Bates, Luke Gubbins.

**Project administration:** Mark Bates, Michael F. Gallagher, Sharon O'Toole, John J. O'Leary.

**Resources:** Mark Bates.

**Software:** Mark Bates, Gordon Blackshields.

**Supervision:** Cathy D. Spillane, Michael F. Gallagher, Amanda McCann, Cara Martin, Sharon O'Toole, John J. O'Leary.

**Validation:** Mark Bates.

**Visualization:** Mark Bates, Sharon O'Toole, John J. O'Leary.

**Writing – original draft:** Mark Bates, Cathy D. Spillane, Michael F. Gallagher, Amanda McCann, Cara Martin, Helen Keegan, Doug Brooks, Sharon O'Toole, John J. O'Leary.

**Writing – review & editing:** Mark Bates, Cathy D. Spillane, Michael F. Gallagher, Amanda McCann, Cara Martin, Helen Keegan, Robert Brooks, Doug Brooks, Stavros Selemidis, Sharon O'Toole, John J. O'Leary.

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
