## [Decision Letter · Decision Letter 0]

27 Nov 2019

PONE-D-19-29743

The MAD2-TLR4-MyD88 Axis

in Paclitaxel Resistance in Ovarian Cancer

PLOS ONE

Dear Dr. Mark Bates,

Thank you for submitting your manuscript to PLOS ONE. After careful consideration, we feel that it has merit but does not fully meet PLOS ONE’s publication criteria as it currently stands. Therefore, we invite you to submit a revised version of the manuscript that addresses the points raised during the review process.

Please particularly pay attention to the use of two non-HGSOC cell lines A2780 and SKOV3 in this study. For others, please revise according to the reviewers' comments.

We would appreciate receiving your revised manuscript by 90 days. To enhance the reproducibility of your results, we recommend that if applicable you deposit your laboratory protocols in protocols.io, where a protocol can be assigned its own identifier (DOI) such that it can be cited independently in the future. For instructions see: http://journals.plos.org/plosone/s/submission-guidelines#loc-laboratory-protocols

We look forward to receiving your revised manuscript.

Kind regards,

David Wai Chan, Ph.D.

Academic Editor

PLOS ONE

Journal Requirements:

2. Please ensure that your Methods section contains a description of Paclitaxel source.

Additional Editor Comments (if provided):

This study reports a novel MAD2-TLR4-MyD88 signaling axis involved in paclitaxel resistance in ovarian cancers. This finding is interesting. However, there are some key findings needed for further strengthen. It's particularly concerned of two cell models A2780 and SKOV3 in this study as they are not HGSOC cell lines. For others, please check for the comments of the reviewers.

Reviewers' comments:

Reviewer's Responses to Questions

**Comments to the Author**

1. Is the manuscript technically sound, and do the data support the conclusions?

Reviewer #1: Yes

Reviewer #2: Yes

Reviewer #3: Yes

2. Has the statistical analysis been performed appropriately and rigorously? 

Reviewer #1: I Don't Know

Reviewer #2: Yes

Reviewer #3: Yes

3. Have the authors made all data underlying the findings in their manuscript fully available?

Reviewer #1: Yes

Reviewer #2: Yes

Reviewer #3: No

4. Is the manuscript presented in an intelligible fashion and written in standard English?

Reviewer #1: Yes

Reviewer #2: Yes

Reviewer #3: Yes

5. Review Comments to the Author

Reviewer #1: The quality of the figures is very poor and the labeling are unreadable, preventing a proper reading and reviewing of the manuscript.

It is just impossible to read the figures in either a printed copy or on computer screen. In a general looking at the graphs and images, the changes are not very small or subtle and thus the conclusions are likely not so significant.

Reviewer #2: This manuscript describes a very well executed series of experiments that have been approached in a comprehensive manner. Both the writing and organization of the presentation are excellent. The authors have been very thorough in how they have addressed their hypothesis. It has been a pleasure reading and reviewing this submission and I strongly support its publication. Especially worthy of inspection by the scientific public are the findings from the micro array analyses suggesting that multiple molecular processes may mitigate sensitivity and contribute to resistance so that singular biomarker dependence may not be sufficient or informative.

I would like the manuscript to address one additional consideration about senescence. It is well-appreciated that borderline tumors of the ovary, granulosa cell tumors and others have very slow growing characteristic which can be equated to some degree with "senescence". These intrinsically slow growing tumors are very non-responsive to chemotherapy. This non-responsiveness arises from a proliferation that is too slow to be halted by chemotherapy. To what extent then can the manipulations described here be the result of the induction of senescence and not specific to the molecular link between TLR4-MyD88 signalling and MAD2? Please address this possibility.

Minor: note that somehow "aswell" rather than "as well" appears multiple times in the manuscript.

Reviewer #3: This study presents data on the MAD2-TLR4-MyD88 axis in paclitaxel resistance in ovarian cancer. The data include, molecular biology experiments showing loss of function and gain of function assays of the three genes and gene expression profiling in loss of function of TLR4 and MAD2 in SKOV3 cells.

Comments to authors.

Results

In general, more attention to details throughout, jargon and abbreviations like - no. of scorers, etc would improve readability of an otherwise well written manuscript. In spite of the document being well written, there are some concerning aspects to the data and rationale for using the approach and reagents.

1. Only 1 siRNA was used to target each gene, hence the variation in knockdown efficiency.

2. SKOV3 and A2780 cells were used as a model of paclitaxel resistance or generic ovarian cancer cells. It is well recognized now that these lines do not represent best models for HGSOC. Evaluating cell lines as tumour models by comparison of genomic profiles (Silvia Domcke et al, Nat. Communications 2013), Characterization of twenty-five ovarian tumour cell lines that phenocopy primary tumours (Tan A. Ince et al., Nat Communications, 2015).

3. Supplementary Figure 1., Section 3, Legend does not correlate with Figure 1C , should be Fig 1D or F?

4. In general the Figure legends are too detailed with general wester blot and other methodologies which should be included in the Methods section.

5. Figure 1E, in general, MAD2 knockdown is not significant, why was this then used in microarray experiments?

6. Figure 3, Section - Suppression of MAD2 induces cellular senescence and paclitaxel resistance.” Authors wrote, 5th line, that MAD2 KD cells exhibited 30% increase in cell viability compare to controls. It appears to me that 30% DECREASED viability as a result of loss of function MAD2, but more viable than treatment alone, indicating that loss of MAD2 improves survival when cells are treated with paclitaxel, which is counter to the hypothesis.

7. Bgal assay is a good representation of the disconnect between proliferation and viability.

8. Curious that cisplatin was mentioned in discussion and no true attention to the platinum based treatments which are truly first line therapy in EOC.

9. The microarray data should be submitted to a public database such as GEO or EMBL.

6. PLOS authors have the option to publish the peer review history of their article (what does this mean?). If published, this will include your full peer review and any attached files.

Reviewer #1: No

Reviewer #2: Yes: Edward John Pavlik

Reviewer #3: No

---

## [Author Response · Author response to Decision Letter 0]

27 Oct 2020

Please see the Response to Reviewers Document

Response to Reviewers

The manuscript has been amended to follow PLOS One style/formatting requirements.

1) The title has been now changed and now uses sentence case with capitilasation only for the first letter of the first word and the names of the proteins MAD2, MyD88 and TLR4

2) Headings now match the font size outlined in the guide and use sentence case

3) The postal code has been removed from the affiliation for authors based in Dublin

4) The supplementary file has been reduced and split up to enhance readability and file naming conventions have been changed to reflect PLOS One policies

2. Please ensure that your Methods section contains a description of Paclitaxel source.

This has now been added to the manuscript in the Methods section PG16 of the manuscript.

“Carboplatin (C2538), Paclitaxel (T402) and DMSO (D2650) were purchased from Sigma Aldrich. Carboplatin was diluted in sterile nuclease free water to a concentration of 10mg/ml and stored at room temperature based on recommendations by the manufacturer. Paclitaxel was diluted in DMSO to a concentration of 50g/l (58.6mM) based on recommendations by the manufacturer aliquoted and stored at -200C while DMSO was kept at room temperature. Aliquots of Carboplatin, Paclitaxel and DMSO were freshly diluted with media for each experiment to the desired working concentrations”

3. PLOS ONE now requires that authors provide the original uncropped and unadjusted images underlying all blot or gel results reported in a submission’s figures or Supporting Information files. This policy and the journal’s other requirements for blot/gel reporting and figure preparation are described in detail at https://journals.plos.org/plosone/s/figures#loc-blot-and-gel-reporting-requirements and https://journals.plos.org/plosone/s/figures#loc-preparing-figures-from-image-files. When you submit your revised manuscript, please ensure that your figures adhere fully to these guidelines and provide the original underlying images for all blot or gel data reported in your submission. See the following link for instructions on providing the original image data: https://journals.plos.org/plosone/s/figures#loc-original-images-for-blots-and-gels. In your cover letter, please note whether your blot/gel image data are in Supporting Information or posted at a public data repository, provide the repository URL if relevant, and provide specific details as to which raw blot/gel images, if any, are not available. Email us at plosone@plos.org if you have any questions.

All uncropped western blots are available in the supplementary material and this is indicated in the figure legend and a note has now also been added to the methods section to further highlight this and it will be noted in the new cover letter. 

4.This study reports a novel MAD2-TLR4-MyD88 signalling axis involved in paclitaxel resistance in ovarian cancers. This finding is interesting. However, there are some key findings needed for further strengthen. It's particularly concerned of two cell models A2780 and SKOV3 in this study as they are not HGSOC cell lines. For others, please check for the comments of the reviewers.

We now include additional MAD2 & TLR4 knockdown results from the serous ovarian cancer cell line OVCAR-3 within the paper. This cell line was chosen as it expresses TLR4, MyD88 and MAD2, other serous lines such as Kuramochi and OAW42 which were also available at our lab do not express TLR4 as indicated by an expression profile comparison which is also now included in the manuscript. Please see the new Figure 3 for these results. Interestingly knockdown of TLR4 in OVCAR-3 cells resulted in a significant increase in MAD2 expression while knockdown of MAD2 in this model resulted in an increase in TLR4 expression similar to the other cell models further highlitghing the relationship between these two biomarkers. The effect of the TLR4 knockdown on MAD2 expression was suprising, although perhaps a partial increase in MAD2 expression was observed post knockdown of TLR4 with both the A2780 and SKOV-3 cells. However this fell below the 2-fold cut-off for consideration and neither were significant. Although significant knockdowns of TLR4 in both cell lines were observed perhaps a certain threshold of TLR4 downregulation is required, notedly with the OVCAR-3, the biological replicate with the highest level of TLR4 knockdown also had the greatest increase in MAD2 expression. 

It is interesting also that ovarian cancer cell lines seem to show stark differences in TLR4 expression, but this is also reflected in ovarian cancer patients. We and others have previously shown that high grade serous ovarian cancer patients which have high TLR4 expression have poorer outcomes. It is possible that within the tumour microenviroment ovarian cancer cells acquire TLR4 through interactions with stromal or immune cells populations something which would not be reflected in cell line models. Additionally the microarray data sets indicate that TLR4 may also highlight aggressive cancer cells undergoing EMT something which might be further amplified in the hypoxic tumour niche as hypoxia is known to promote the emergence of stem like phenotypes and can also induce the downregulation of MAD2 as we have previously shown (Prencipe 2010, PMID20676051) another factor which may not be observed in conventional 2D cell models.

Fig 3. Cell line cross comparison and OVCAR-3 TLR4 and MAD2 siRNA knockdown. (A) MyD88, TLR4 and MAD2 gene expression in A2780, OVCAR-3, PEO1, OAW42, KURAMOCHI and 59M cells relative to SKOV-3 cells. (B) MyD88, TLR4 and MAD2 gene expression in OVCAR-3 cells following siRNA knockdown of TLR4 or MAD2. Gene expression levels were normalised to the endogenous control GAPDH and calibrated to that of untreated cells to establish the relative change in gene expression .

Preliminary data (n=1) below from PEO1 cells an additional serous ovarian cell line indicate that this effect may not be universal however, although this particular cell line interestingly has a P16 Null phenotype (Furlong 2009, PMID: 22069160) and may not be capable of undergoing cellular senescence, this is something we hope to explore further in future studies.

Furthermore, knockdown of TLR4 has been previously shown to enhance sensitivity to paclitaxel in the OVCAR-3 cell line (Szajnik 2009, PMID 19826413) similar to what we have observed in the SKOV-3 cell line. Additionally, we have also demonstrated the senescence effect following knockdown of MAD2 in MCF-7 breast cancer cells (Prencipe 2009, PMID: 2788249) demonstrating that this is likely not restricted to specific subtypes of ovarian cancer nor is it even restricted to ovarian cancer but it is probably a ubiquitous phenomenon which likely occurs in various cancers. A link between MAD2 and senescence has also been demonstrated in prostate cancer (To-Ho 2007, PMID 17621272). Also, paclitaxel most importantly is a known ligand for TLR4 (Byrd-Leifer 2001, PMID: 11500829) and the effect of TLR4 on paclitaxel resistance similarly is not restricted to ovarian cancer either, please see Rajput 2013, PMID 23720768 & Kashani 2020 PMID: s12026-019-09113-8). 

5. Is the manuscript technically sound, and do the data support the conclusions? The manuscript must describe a technically sound piece of scientific research with data that supports the conclusions. Experiments must have been conducted rigorously, with appropriate controls, replication, and sample sizes. The conclusions must be drawn appropriately based on the data presented.

Reviewer #1: Yes

Reviewer #2: Yes

Reviewer #3: Yes

6. Has the statistical analysis been performed appropriately and rigorously?

Reviewer #1: I Don't Know

Reviewer #2: Yes

Reviewer #3: Yes

7. Have the authors made all data underlying the findings in their manuscript fully available?

Reviewer #1: Yes

Reviewer #2: Yes

Reviewer #3: No

 All data is available in the manuscript, supporting material or otherwise where indicated in the data availability statement

8. Is the manuscript presented in an intelligible fashion and written in standard English? PLOS ONE does not copyedit accepted manuscripts, so the language in submitted articles must be clear, correct, and unambiguous. Any typographical or grammatical errors should be corrected at revision, so please note any specific errors here. 

Reviewer #1: Yes

Reviewer #2: Yes

Reviewer #3: Yes

9. Reviewer #1: The quality of the figures is very poor and the labeling are unreadable, preventing a proper reading and reviewing of the manuscript. It is just impossible to read the figures in either a printed copy or on computer screen. In a general looking at the graphs and images, the changes are not very small or subtle and thus the conclusions are likely not so significant.

Apologies there appears to have been some compression issue with images and this has been amended in the revised version of the manuscript.

10.Reviewer #2: This manuscript describes a very well executed series of experiments that have been approached in a comprehensive manner. Both the writing and organization of the presentation are excellent. The authors have been very thorough in how they have addressed their hypothesis. It has been a pleasure reading and reviewing this submission and I strongly support its publication. Especially worthy of inspection by the scientific public are the findings from the micro array analyses suggesting that multiple molecular processes may mitigate sensitivity and contribute to resistance so that singular biomarker dependence may not be sufficient or informative. I would like the manuscript to address one additional consideration about senescence. It is well-appreciated that borderline tumors of the ovary, granulosa cell tumors and others have very slow growing characteristic which can be equated to some degree with "senescence". These intrinsically slow growing tumors are very non-responsive to chemotherapy. This non-responsiveness arises from a proliferation that is too slow to be halted by chemotherapy. To what extent then can the manipulations described here be the result of the induction of senescence and not specific to the molecular link between TLR4-MyD88 signalling and MAD2? Please address this possibility.

I would like to thank reviewer #2, for their comment I think they make a very interesting point. It is perhaps a difficult question to answer, it seems as though there is a synergy between MAD2 and the TLR4 signalling pathway. Whether senescence can occur without the TLR4 pathway it would be interesting to determine, something we may be able to answer through further exploration with TLR4 negative cell lines. MyD88 itself does not appear to be required as the senescence effect as a result of MAD2 suppression has been shown by our group to occur in both SKOV-3 cells (MyD88 positive) and A2780 (MyD88 Negative, See Furlong 2009, PMID: 22069160). Certainly, you can have the TLR4 pathway mechanism of paclitaxel resistance on its own which is driving apoptotic resistance and pro-tumorigenic inflammatory niche formation. Whether MAD2 suppression can lead to paclitaxel resistance in the absence of senescence I suppose we can’t answer currently, although for the purpose of this article we are attributing the effect of MAD2 suppression on paclitaxel resistance to senescence. It would be interesting though to interrogate this further in cell models which have functional defects which don’t allow them to undergo senescence. 

A big debate around senescence also is whether the process is reversible “in vivo”, can these cells recover after a hibernation phase and re-emerge once the chemotherapy subsides. The 2nd thing with this is whether all cells undergo senescence in that type of microenvironment, even in this model not all off the cells display an increase in size and display enhanced B-gal activity, although this could reflect transfection efficiency. 

We do describe in a recent a review in Cancer Letters (PMID: 31593803) that perhaps these senescent cells act as a physiological barrier and soak up the paclitaxel essentially and facilitate the emergence/re-emergence of non-senescent tumour cell populations such as cancer stem cells, which are known to display many of the features described in the TLR4 microarray data set presented in the paper. Thus, we really need to target these senescent cell populations.

11.Minor: note that somehow "aswell" rather than "as well" appears multiple times in the manuscript.

This error has now been amended.

12.Reviewer #3: This study presents data on the MAD2-TLR4-MyD88 axis in paclitaxel resistance in ovarian cancer. The data include molecular biology experiments showing loss of function and gain of function assays of the three genes and gene expression profiling in loss of function of TLR4 and MAD2 in SKOV3 cells. In general, more attention to details throughout, jargon and abbreviations like - no. of scorers, etc would improve readability of an otherwise well written manuscript. In spite of the document being well written, there are some concerning aspects to the data and rationale for using the approach and reagents.

The manuscript has now been amended to address this

13. Only 1 siRNA was used to target each gene, hence the variation in knockdown efficiency.

The TLR4 siRNA is a silencer select siRNA from Thermofisher. This siRNA has been chemically modified with a Locked Nucleic Acid (LNA) a method which has been shown to reduce many undesired, sequence-related off-target effects (PMID: 15653644). Furthermore, Thermofisher measure potential off-target activity using microarray analysis and bioinformatically screen their siRNAs to maximise accuracy and specificity. This is a highly validated commercial siRNA from a well-established supplier.

Furthermore, this siRNA has been validated in a number of publications including high impact journals such as Nature Communications. See below

PMID: 24614850

PMID: 28832545

We also previously published an article in PLOS one on the effect of this particular siRNA on paclitaxel sensitivity PMID: 24977712. The work presented in this study is essentially a follow-on study exploring this mechanism further.

We do agree that siRNA pools, shRNA or even other technologies that didn’t exist when this study began such as CRISPR might lead to better targeting of this gene and reduce any potential off-target effects even further. However, regardless of the mechanism of interrogation, we still feel the results presented here are valid and are based on a confirmed significant reduction of TLR4 protein/gene expression.

This study sought however to confirm two things, first did the knockdown of TLR4 and the resulting impact on paclitaxel sensitivity influence MAD2 expression and secondly, we wanted to further interrogate this pathway using microarray technology, again with a particular focus on the SKOV-3 cell line.

As for the MAD2 knockdown, this was performed using siRNA pools specifically a SMARTpool siRNA purchased from Dharmacon and for this experiment, it is a non-issue. Additionally, in either knockdown, a scrambled or negative control siRNA was used to demonstrate any off-target effects of activating the RNAi machinery.

14. SKOV3 and A2780 cells were used as a model of paclitaxel resistance or generic ovarian cancer cells. It is well recognized now that these lines do not represent best models for HGSOC. Evaluating cell lines as tumour models by comparison of genomic profiles (Silvia Domcke et al, Nat. Communications 2013), Characterization of twenty-five ovarian tumour cell lines that phenocopy primary tumours (Tan A. Ince et al., Nat Communications, 2015).

In 2013, partway through this project, the Domcke et al. (2013) study questioned the appropriateness of many cell models for ovarian cancer research. The article highlighted how many ovarian cancer cell lines based on genome sequencing and mutational analysis may not be truly representative of high grade serous ovarian cancer. Although this research article does highlight a very important issue in ovarian cancer research, it must also be acknowledged that cell models are simply models of disease and are only representative of a single ovarian cancer patient. 

The A2780 and SKOV-3 cells models, were the most frequently utilised cell models for ovarian cancer research and are still widely utilised today. The SKOV-3 cell model is among one of the cell models used in the national cancer institute’s NCI-60 panel used to test new cytotoxic drugs by the FDA. The two cell models A2780 and SKOV-3 were also chosen mainly for the fact that they represented positive and negative models of MyD88 respectively, with the A2780 model being MyD88 null and paclitaxel sensitive and the SKOV-3 cells being MyD88 positive and paclitaxel resistant. Both also expressed MAD2 and knockdown of MAD2 in the A2780 cells had been shown to induce paclitaxel resistance (PMID: 22069160, PMID: 21063845). They also both express TLR4, however, only knockdown of TLR4 in SKOV-3 cells but not A2780 cells had been shown to restore paclitaxel sensitivity [PMID: 19826413, PMID: 24527095]. Therefore, we felt these cells were still appropriate models to take forward for evaluation. While we appreciate HGS is the most common subtype, we do think there is merit in this approach even for other ovarian cancer subtypes.

In the Domcke et al study, sequencing data performed on cell lines was compared to TCGA data. While the TCGA has given an enormous amount of data to scientists, there are some concerns over the interpretation of this data which have preventing the translation of the 4 molecular subtypes of HGSOC identified from the TCGA data into the clinic. Criticisms of the TGCA data are published here [PMID:18698038, PMID:21720365] which include technical limitations such as inconsistency between platforms, sample batch effects [PMID:20838408], reagent batch effects and inconsistent macro-dissection resulting in variable stroma to tumour content in the gene expression data [PMID:18698038, PMID:21720365]. Our own work in which A2780 were injected into mouse models, tumours formed which histologically resembled high grade serous ovarian tumours as confirmed by a pathologist. 

Furthermore, we have assessed MAD2, TLR4 and MyD88 to be prognostic in patients from high grade serous ovarian tumours. But this paper goes beyond the use of single biomarkers and looks at more complex pathways at play many of which are known to contribute to ovarian cancer pathogenesis.

We have also previously demonstrated the senescence effect in breast cancer cell models, this may, in fact, be a universal phenomenon, The effect of TLR4 suppression on paclitaxel sensitivity has also been demonstrated previously in breast and other tumours.

We have also now included data on the OVCAR-3 serous ovarian cell line, which can be observed above.

Results Section – PG5/6 of the manuscript – “To further explore the link between TLR4 and MAD2 we next analysed the expression of MAD2, TLR4 and MyD88 in 5 additional ovarian cancer cell lines; OVCAR-3, PEO1, OAW42, KURAMOCHI and 59M cells (Fig 3A). Of these only OVCAR-3 and PEO1 expressed TLR4, MyD88 and MAD2. OAW42, KURAMOCHI and 59M were TLR4 negative. OVCAR-3 cells due to their TLR4 positivity and as a representative model of serous ovarian cancer were subsequently transfected with siRNA targeting TLR4 or MAD2 and then TLR4, MAD2 and MyD88 expression levels were assessed. Interestingly knockdown of TLR4 or MAD2 in the OVCAR-3 cell model caused a significant 2.4 and 2.9 fold increase in MAD2 or TLR4 expression respectively further highlighting an important link between these two biomarkers (Fig 3B).”

Discussion PG10 of the manuscript - “Intererestingly however in the OVCAR-3 serous ovarian model a significant increase in MAD2 expression was observed. One observation with this experiment was that the replicate with the highest knockdown of TLR4 had the highest upregulation of MAD2”, “Furthermore, suppression of MAD2 expression using siRNA in A2780, OVCAR-3 and SKOV-3 cell lines led to a significant increase in TLR4 gene expression levels demonstrating a key link between TLR4 and MAD2”

15. Supplementary Figure 1., Section 3, Legend does not correlate with Figure 1C , should be Fig 1D or F?

This error has been amended

16. In general the Figure legends are too detailed with general wester blot and other methodologies which should be included in the Methods section.

The amount of detail in the figure legends has been reduced in both the main text and supplementary data.

17. Figure 1E, in general, MAD2 knockdown is not significant, why was this then used in microarray experiments?

Apologies there seems to have been some compression issue with the images that were uploaded. Figure 1E however is examining MAD2 gene expression following knockdown of either TLR4 or MyD88 (Both of which were significant), this did not affect MAD2 gene expression. This experiment was important for assessing the relationship between these 3 biomarkers. The TLR4 knockdown was taken forward for microarray analysis as this induced enhanced paclitaxel sensitivity in the SKOV-3 cell model but not knockdown of MyD88. 

The MAD2 knockdown is depicted in Figure 2 and was highly significant in both SKOV-3 and A2780 cells, the SKOV-3 cell was subsequently chosen for microarrays being a MyD88 positive cell line, which had been shown to display both mechanisms of paclitaxel resistance/sensitivity independently. 

18. Figure 3, Section - Suppression of MAD2 induces cellular senescence and paclitaxel resistance.” Authors wrote, 5th line, that MAD2 KD cells exhibited 30% increase in cell viability compare to controls. It appears to me that 30% DECREASED viability as a result of loss of function MAD2, but more viable than treatment alone, indicating that loss of MAD2 improves survival when cells are treated with paclitaxel, which is counter to the hypothesis.

I believe the reviewer is referring to below text

“When SKOV-3 cells were treated with a lethal 1µM dose of paclitaxel following knockdown of MAD2, they exhibited a 30% increase in cell viability compared to untreated cells and scrambled negative control cells treated with the same dose of paclitaxel demonstrating that SKOV-3 cells were rendered paclitaxel-resistant (Fig 3B)”

MAD2 knockdown cells do indeed exhibit a decrease in viability according to the CCK-8 assay by about 30% compared to untreated/untransfected cells which did not receive any drug. Although very little cell loss was observed post knockdown of MAD2, any decrease here in “viability” as determined by the cell viability assay in these cells is likely rather a representation of decreased cell proliferation rates due to the induction of senescence. 

However, compared to untransfected cells or cells transfected with the negative control siRNA viability is enhanced by 30% in the MAD2 knockdown cells definitively demonstrating that these cells are paclitaxel-resistant which is in line with findings in our previous works. It is not counter to our hypothesis, the fact that loss of MAD2 improves the survival of cells treated with paclitaxel is exactly our hypothesis and exactly what we have shown, and this is also mirrored in patients with high grade serous ovarian cancer i.e. patients with low expression of MAD2 have worse outcomes (Furlong 2009, PMID: 22069160).

The wording of this section (Pg 6 of the manuscript) has been amended to enhance readability.

“Suppression of MAD2 induces cellular senescence and paclitaxel resistance

Following knockdown of MAD2, SKOV-3 cells exhibited an increase in cell and nuclear size and alteration of cell shape (Fig 4A). Subsequently, the chemoresponse of SKOV-3 cells to paclitaxel following knockdown of MAD2 was assessed (Fig 4C). When SKOV-3 cells were treated with a 20nM or 1µM dose of paclitaxel following knockdown of MAD2, they exhibited a reduction in cell viability of 36.2% and 36.1% compared with untransfected cells which were not treated with paclitaxel. In contrast, untransfected cells or cells transfected with the scrambled negative control which were treated with 20nM of paclitaxel exhibited a decrease in cell viability of 55.4% and 56.3% respectively. While untransfected cells or cells transfected with the scrambled negative control siRNA which were treated with 1µM of paclitaxel exhibited a decrease in cell viability of 66.2% and 66.4% respectively. Additionally, despite there being no visual signs of cytotoxicity, untreated cells transfected with MAD2 siRNA also exhibited a 19% significant reduction in cell viability compared to untransfected cells which were untreated potentially indicating a reduction in cell proliferation. Furthermore, in transfected cells treated with either dose of paclitaxel minimal if any visual signs of cytotoxicity were observed. In fact, the addition of paclitaxel appeared to accelerate the timeframe for the emergence of the enlarged cell phenotype. Additionally, the difference in cell viability between cells transfected with MAD2 siRNA which were untreated and transfected cells treated with either dose of paclitaxel was not statistically significant. This result was further supported by a trypan blue exclusion assay which detected a significant decrease in the number of dead cells in the supernatants of paclitaxel treated cells following knockdown of MAD2 compared to controls (Fig 4E). The results indicated that SKOV-3 cells transfected with siRNA targeting MAD2 were rendered resistant to paclitaxel and potentially undergoing cellular senescence.”

CCK8 Assay- Now Figure 4C, PG22 in the manuscript and also now includes and additional dose 20uM to match what was examined with the Trypan blue exclusion assay.

We also now include a previous trypan blue exclusion assay we performed on the supernatants of cells transfected and treated with paclitaxel which also reflects this.

Trypan Blue Exclusion Assay Fig 4E, PG22

Details of the trypan blue exclusion assay experiment are now also included in the methods section pg18

“Trypan Blue dye exclusion assay

SKOV-3 cells were transfected into 6 well plates with siRNA targeting MAD2, a non-targeting scrambled negative control siRNA or were left untreated for 72 hours. After 72 hours, cells were left untreated, treated with DMSO or were treated with 20nM paclitaxel and incubated for a further 48 hours. After the 48-hour drug incubation time, Supernatants from each well were collected and wells rinsed with PBS to remove any residual dying cells. Collected supernatant and washings were pelleted by centrifugation and resuspended in a small volume of PBS. Cell suspensions were then mixed at 1:1 ratio with trypan blue (T8154, Sigma Aldrich) and the number of dead cells were counted using a haemocytometer”

The fact that knockdown of MAD2 increases TLR4 expression is also curious given that we previously published that knockdown of TLR4 enhances the sensitivity of SKOV-3 cells to paclitaxel (d’Adhemar 2014, PMID 24977712)

Interestingly the addition of paclitaxel actually also seemed to accelerate the emergence of the senescent phenotype observed at 120 hours. In essence paclitaxel not only doesn’t kill these MAD2 knockdown cells, it actually amplifies the senescence phenotype (Therapy induced senescence is a frequently described phenomenon). As stated below the Bgal assay also further demonstrates the disconnect between proliferation and viability.

19. Bgal assay is a good representation of the disconnect between proliferation and viability.

We agree with the reviewer on this, a limitation of cell viability assays is that they are representative of cell numbers rather than mechanistic/visual effects

20. Curious that cisplatin was mentioned in discussion and no true attention to the platinum-based treatments which are truly first line therapy in EOC.

We now include new results on this. Neither the knockdown of TLR4 or MAD2 had any impact on carboplatin sensitivity in the SKOV-3 model. The effect of knocking down TLR4 in SKOV-3 cells on carboplatin sensitivity is line with a previous report (Szajnik 2009, PMID 19826413) Previous work by our group (Unpublished data) had shown a similar effect with A2780 cells with cisplatin. This demonstrates that the effects with TLR4 and MAD2 seem to be specific to paclitaxel and this is now also discussed. 

SKOV-3 cells were treated with a 200uM dose of paclitaxel now Figure 4F

CCK-8 assay results for SKOV-3 cells treated with 200µM carboplatin for 48 hours following a 24-hour transfection. Results are expressed as mean +/-SD, n=3. *p<0.05, **p<0.01, ***p<0.001 (Student’s t-test).

This was based on IC50 results with the SKOV-3 cells, see below which is also now included in supplementary data.

SKOV-3 carboplatin dose-response curve. SKOV-3 cells were treated with various doses of carboplatin for 48 hours. Cell viability was assessed using the cell counting kit 8 (CCK-8). The IC50 value for SKOV-3 at this timepoint was 139.1µM.

Result Section PG7 of the manuscript- “We also assessed the cytotoxicity of SKOV-3 cells to carboplatin following knockdown of either TLR4 or MAD2. Cells were transfected for 24 hours and then treated with a 200µM dose of carboplatin or left untreated, however, neither knockdown of TLR4 or MAD2 altered the response of SKOV-3 cells to carboplatin (Fig 4F). 

Discussion Section PG12 of the manuscript - “The results also demonstrate that these senescent cell populations appear to be selectively resistant to paclitaxel but are sensitive to carboplatin, therefore patients could potentially be selected out for single arm therapy with carboplatin”

21. The microarray data should be submitted to a public database such as GEO or EMBL.

The Affymetrix microarray data sets generated as part of this study are available in an ArrayExpress repository, accession #370077 as indicated in the data availability statement.

22. PLOS authors have the option to publish the peer review history of their article (what does this mean?). If published, this will include your full peer review and any attached files. Yes, we are happy for the peer review history to be published.

---

## [Decision Letter · Decision Letter 1]

26 Nov 2020

The role of the MAD2-TLR4-MyD88 axis

in paclitaxel resistance in ovarian cancer

PONE-D-19-29743R1

Dear Dr. Bates,

We’re pleased to inform you that your manuscript has been judged scientifically suitable for publication and will be formally accepted for publication once it meets all outstanding technical requirements.

Kind regards,

David Wai Chan, Ph.D.

Academic Editor

PLOS ONE

Additional Editor Comments (optional):

Reviewers' comments:

Reviewer's Responses to Questions

**Comments to the Author**

1. If the authors have adequately addressed your comments raised in a previous round of review and you feel that this manuscript is now acceptable for publication, you may indicate that here to bypass the “Comments to the Author” section, enter your conflict of interest statement in the “Confidential to Editor” section, and submit your "Accept" recommendation.

Reviewer #1: (No Response)

Reviewer #3: All comments have been addressed

2. Is the manuscript technically sound, and do the data support the conclusions?

Reviewer #1: Partly

Reviewer #3: Yes

3. Has the statistical analysis been performed appropriately and rigorously? 

Reviewer #1: Yes

Reviewer #3: Yes

4. Have the authors made all data underlying the findings in their manuscript fully available?

Reviewer #1: Yes

Reviewer #3: Yes

5. Is the manuscript presented in an intelligible fashion and written in standard English?

Reviewer #1: Yes

Reviewer #3: Yes

6. Review Comments to the Author

Reviewer #1: The presentation is improved, and the authors appear to make good efforts to change/improve the manuscript.

However, the mechanistic links between the markers are not investigated, and the changes/impacts of each gene modulation are small or moderate. The study provides little new understanding on paclitaxel resistance.

Such manuscript may be publishable but is a low quality study.

Reviewer #3: The authors have improved this manuscript significantly, with inclusion of other cancer cell lines and responded to the reviewers comments.

7. PLOS authors have the option to publish the peer review history of their article (what does this mean?). If published, this will include your full peer review and any attached files.

Reviewer #1: No

Reviewer #3: No

---

## [Editor Report · Acceptance letter]

14 Dec 2020

PONE-D-19-29743R1 

The role of the MAD2-TLR4-MyD88 axis in paclitaxel resistance in ovarian cancer 

Dear Dr. Bates:

I'm pleased to inform you that your manuscript has been deemed suitable for publication in PLOS ONE. Congratulations! Your manuscript is now with our production department. 

Kind regards, 

on behalf of

Dr. David Wai Chan 

Academic Editor

PLOS ONE